# A Correlational Study of Two U.S. State Extension Professionals’ Behavioral Intentions to Improve Sustainable Food Chains through Precision Farming Practices

**DOI:** 10.3390/foods12112208

**Published:** 2023-05-31

**Authors:** Chin-Ling Lee, Robert Strong, Gary Briers, Theresa Murphrey, Nithya Rajan, Shelli Rampold

**Affiliations:** 1Taoyuan District Agricultural Research and Extension Station, Council of Agriculture, Taoyuan 32745, Taiwan; chinling@tydais.gov.tw; 2Department of Agricultural Leadership, Education, and Communications, Texas A&M University, College Station, TX 77843, USA; g-briers@tamu.edu (G.B.); t-murphrey@tamu.edu (T.M.); 3Department of Soil and Crop Sciences, Texas A&M University, College Station, TX 77843, USA; nrajan@tamu.edu; 4Department of Agricultural Leadership, Education, and Communications, The University of Tennessee, Knoxville, TN 37996, USA; srampold@utk.edu

**Keywords:** innovative agricultural practices, UTAUT model, performance expectancy, social influence, professional development programs, change agents

## Abstract

Precision farming provides one of the most important solutions for managing agricultural production to advance global food security. Extending professionals’ competencies to promote precision farming practices can increase the adoption rate, ultimately impacting food security. Many studies have addressed barriers to the adoption of precision farming technologies from the farmers’ perspective. However, few are available data on the perspectives of extension professionals. Agricultural extension professionals play an important role in innovative agricultural technology adoption. Thus, this study applied four constructs from the unified theory of acceptance and use of technology (UTAUT) model to investigate behavioral intentions to promote precision farming among extension professionals from two extension systems. In total, 102 (*N* = 102) agricultural extension professionals were surveyed. The results indicated that performance expectancy and social influence were individually significant predictors of extension professional behavioral intentions to promote precision farming technologies. There were no significant differences between the professionals of two extension systems. Gender, age, and years of service did not affect extension professionals’ intention to promote precision agriculture technologies. The data suggested the need for training programs to develop advanced competencies to promote agricultural innovation. This study contributes to the future professional development programs for extension professionals on communicating innovations to address food security and sustainability issues.

## 1. Introduction

Food security is an important issue worldwide. According to [1], the FAO anticipated the depreciation of food security by 2030. As the global population grows toward 9.6 billion by 2050 [2], many solutions have been proposed to meet future food needs while maintaining the environment [3]. Many challenges, such as extreme climates, threaten food security [4] and require global scientific innovation focusing on sustainable agricultural practices that support healthy agricultural dietary solutions [5]. Food experts indicated that while there is no single solution to food security, technology can play an important role in global food security and in maintaining agricultural sustainability [6]. Currently, agriculture has further opportunities to apply more intelligent tools because of the widespread use of digital technologies in various fields [7]. The advances in information and communication technologies have enabled farmers to access a large amount of site-specific data for their farms, which provide farmers with the opportunity to apply precision-farming approaches to make more accurate decisions to reduce unnecessary inputs and achieve production quantity and quality [8,9]. Thus, precision-farming technologies have received increased attention in providing solutions for food production to feed a growing global population [10,11,12,13].

Precision agriculture is the application of information technology to manage agricultural production to better advance global food security [11]. In addition, adopting precision agriculture technologies is necessary to adjust to [14] extreme climates that can bring new pests and diseases [15]. However, studies have noted that precision farming technologies require a variety of knowledge and skills among farmers and that the application of these technologies on farms may change farmers’ approach to farm management from an experience-driven to a data-driven practice [16,17].

Extension professionals serve as agricultural change agents that represent an organization in local communities by interfacing with community leaders and stakeholders [18]. The extent farmers observe members of their social system using precision-faming applications influences the cultural structure (change agents and opinion leaders of farmers [19]. Farmer are more willing to use an innovation that is promoted by an opinion leader in their social system versus one that is not. However, the difficulties in accessing the technology and extension service create barriers to adopting agricultural technology [20]. The adoption of precision farming technologies depends much on the extension’s dissemination of each innovation’s advantages [21]. Agricultural extension change agents are a factor that influences the decision-making process of farmers to adopt or reject innovations [22]. Lee et al. [23] found that extension professionals in the extension system lacked professional development in promoting the adoption of precision farming technologies by farmers. Lack of information dissemination and extension service will result in a low precision farming technologies adoption rate [24]. Emmanuel et al. [25] suggested that various institutions should train more extension professionals because they have significant impact on agricultural innovative technology adoption by farmers. Therefore, developing extension professionals’ competency to enhance production practices with precision farming technologies is a necessary part of improving services that meet the needs of farmers [26]. Many studies have addressed barriers to adoption of precision-farming technologies, including accessibility, socioeconomic status (e.g., age, gender, education), digital divide, misinformation, and availability [27]. However, there are few data on extension professionals’ behavioral intention to promote precision farming technologies to increase the adoption rate by farmers. In addition, due to the nature of the precision farming technology, adoption and diffusion vary by crop variety, region, and country [28]. Thus, this study’s purpose was (1) to explore the perspectives of agricultural extension professionals from two extension systems in the United States and (2) to identify the factors that influence their intentions to promote precision farming technology practices. The contribution of this study sought to improve precision farming information dissemination, extension professional development needs, and approaches to improve farmer adoption of precision farming innovations that optimized food chain and land sustainability.

## 2. Theoretical Framework

The unified theory of acceptance and use of technology (UTAUT) model by Venkatesh et al. [29] was used to investigate the behavioral intentions of agricultural extension professionals, including county agents and state specialists, to promote precision farming technologies in their extension roles. The UTAUT model combines eight prior models or theories to explore technology adoption [29]. Many studies indicated that the UTAUT variables applied to determine users’ technology adoption vary in different contexts [30]. Many studies applying the UTAUT model in an agricultural context have focused on technology adoption by end users (i.e., farmers) [31], not by promotors. The UTAUT model has proved to be an appropriate contextualization tool to measure the adoption of agricultural technologies [32,33]. Four main constructs play important roles as variables influencing individuals’ acceptance behavior: performance, effort expectancy, social influence, and facilitating conditions.

Performance expectancy refers to the degree to which an individual believes using technologies improves performance [34]. Viewing the relevant models integrated into the UTAUT model, performance expectancy echoes the perceived usefulness of technology acceptance model [35] and relative advantage of innovation of diffusion theory [22].

Effort expectancy is the perception of ease of use. If individuals feel they don’t need to make a huge effort to use technology, their willingness to use it will increase [36].

Social influence is the extent to which individuals believe that people who are important to them think they should use technology. Prior studies have indicated that social influence can affect individuals’ behavior [37].

Facilitating conditions refer to the extent to which an individual believes that infrastructure to support the use of the technology exists. Studies have demonstrated that the higher the facilitating conditions provided, the greater the chances of technology adoption by end users [38,39].

Venkatesh et al. [29] identified four moderators—age, gender, voluntariness, and experience—that may affect the relationship between four key determining constructs and intention of innovation usage. Existing UTAUT studies supported age, gender, and experience as moderators in innovation adoption [40,41,42]. However, expectations about an individual to engage in precision farming technology promotion and adoption may change through age and previous work experience [43]. Thus, in this study, we modified the UTAUT model to include age, gender, and experience as independent variables instead of moderating variables. We used participants’ years of service in extension positions of the participants as a variable of experience. The modified UTAUT model is shown in Figure 1.

This study used this conceptual model to examine whether selected factors of extension professionals affect their behavioral intentions to promote precision farming technologies. Specifically, the objectives of this study were to:(1)Describe agricultural extension professionals’ performance expectancy, effort expectancy, social influence, facilitating conditions, and behavioral intentions to promote precision farming technologies.(2)Investigate the relationship between performance expectancy, effort expectancy, social influence, and facilitating conditions on agricultural extension professionals’ behavioral intentions to promote precision farming technologies.(3)Examine the mean difference of four variables—performance expectancy, effort expectancy, social influence, and facilitating condition—between two extension system groups.(4)Predict behavioral intentions using independent variables (performance expectancy, effort expectancy, social influence, facilitating conditions, age, gender, and years of service).

## 3. Method

### 3.1. Population and Samples

The technology that can be applied to precision farming varies by crop and region [28]. Therefore, to have a more complete understanding the perspectives of agricultural Extension professionals in various fields toward the promotion of precision farming, this study selected two extension systems—the University of California’s Cooperative Extension (UC Extension) and the University of Tennessee Extension (UT Extension)—in the United States and conducted a survey design with an instrument distributed to 468 agricultural extension professionals in the UC and UT extension systems. The research procedures and instrument were approved by Texas A&M University under IRB 2022-0175M for the implementation of the study with these two extension systems.

The population comprised individuals identified as extension agents or specialists who may actively or possibly work with precision farming technologies with crop producers in their role in either the UC or UT extension systems. We used a list of UT extension professionals obtained from the University of Tennessee Extension Department and a list of California Extension professionals identified by the first author as the sampling frame. Random sampling was used to select participants from the sampling frame [44]. The sample consisted of 102 agents and specialists—51 from UT Extension and 51 from UC Extension.

Food crops were the primary crop among extension professionals who responded to the survey—32 of 37 UC extension professionals (*n* = 32, *N* = 37; 86%) and 39 of 46 UT extension professionals (*n* = 39, *N* = 46; 85%) responded with crop varieties they worked with. The extension professionals in the two extension systems differed in the types of crops they specialized in within the food crop category. According to the crop variety answered by the participants, the food crop category included vegetables (i.e., leafy green, onion, lettuce, etc.), fruits (i.e., watermelon, berries, etc.), and grains (i.e., rice, wheat, etc.). In total, 29 of 46 UT Extension professionals (*n* = 26, *N* = 46; 52%) reported that grains were the dominant food crops they specialized in and that fruits (*n* = 10, *N* = 46; 22%) were the least-reported crop for their work. UC extension professionals’ responses in terms of crops were the opposite of UT’s responses, with fruit reported as the primary crops in which extension professionals worked—with 25 of 37 UC Extension professionals responding (*n* = 25, *N* = 37; 68%)—while grains (*n* = 5, *N* = 37; 14%) were the least-reported crop varieties in which they specialized. Figure 2 depicted the types of crops in which the surveyed extension professionals from two extension systems were specialized.

### 3.2. Data Collection

A questionnaire was developed to operationalize and measure five constructs important in this study (performance expectancy, effort expectancy, facilitating conditions, social influence, and behavioral intention to promote precision agriculture technologies). Each construct was measured using a multi-item scale. To measure participants’ level of agreement with the provided statements for performance expectancy, effort expectancy, facilitating conditions, social influence, and behavioral intention constructs, we used a 5-point response scale of 1 = *strongly disagree*, 2 = *disagree*, 3 = *neither agree nor disagree*, 4 = *agree*, and 5 = *strongly agree*. The statements used to measure each construct were based on the literature [45] and were adjusted slightly to be appropriate for research objectives. We developed the questionnaire in the Qualtrics survey platform, and we administered the questionnaire via email. We followed Dillman et al.’s [46] five-step survey design method to contact participants and collect data.

A total of 102 participants accessed the instrument with a response rate of 22% (*n* = 102, *N* = 468). In total, 71 out of 102 participants responded with their gender, and more than half of the participants who answered the gender question were male (*n* = 51, *f* = 68.92%). We used a cross-sectional design in which we tested age differences based on U.S. Census age groups between young (<34 years), middle-aged (35 to 54 years), and older (>55 years) participants. In addition, we tested the experience variable using groups of lesser (<10 years), moderate (10 to 20 years), and high (>20 years) experience. In total, 72 out of 102 participants responded with their age and years of service. The age range of participants was 25 to 75 years, with a mean age of 49 years old (*n* = 72, *SD* = 14.95). The participants’ years of service as an extension agent or specialist ranged from less than 1 year to 49 years, with a mean of 17.59 years of service (*n* = 72, *SD* = 14.21). Table 1 summarizes some of the demographic characteristics of the sample.

### 3.3. Data Analysis

We used SPSS 28.0 to analyze the data collected through the Qualtrics survey platform. Descriptive and inferential statistics were used to analyze data [47]. Cronbach’s [48] alpha coefficients were calculated to measure the internal consistency of five constructs of this study, yielding coefficients of 0.93 for performance expectancy, 0.90 for effort expectancy, 0.72 for facilitating condition, 0.84 for social influence, and 0.96 for behavioral intention. In addition, Cronbach [48] indicated that reliability coefficients of 0.70 or higher are acceptable and that those of 0.80 or higher are good.

Davis’ [49] conventions were used to describe the magnitude of the correlation between pairs of variables: 0.01 ≥ *r* ≥ 0.09 = *Negligible*, 0.10 ≥ *r* ≥ 0.29 = *Low*, 0.30 ≥ *r* ≥ 0.49 = *Moderate*, 0.50 ≥ *r* ≥ 0.69 = *Substantial*, *r* ≥ 0.70 = *Very Strong*. ANOVA analysis and *t*-test were used to test the group mean differences of the significant independent variables. A Pearson correlation analysis was used to examine the relationship between constructs used in the UTAUT model. The multiple linear regression analysis was used to test whether four constructs (performance expectancy, effort expectancy, social influence, and facilitating conditions) have statistically significant power to predict the dependent variable, behavioral intention to promote precision-farming technologies. The regression model used in this study is shown below.
Y = β_0_ + β_1_X_1_ + β_2_X_2_ +β_3_X_3_ + β_4_X_4_ + e

Y = Dependent variable

β_0_ = Population Y intercept

β_i_ = Population Slope Coefficient

Xi = Independent Variable

e = Random error

## 4. Results

### 4.1. Descriptive Results

The first objective of this study was to describe agricultural extension professionals’ performance expectancy, effort expectancy, social influence, facilitating conditions, and behavioral intentions to promote precision farming technologies. Data collected from the Tennessee extension system showed that the highest-scoring construct was “Behavioral Intention” (*M* = 3.36, *SD* = 0.72); the lowest-scoring construct was “Facilitating Conditions” (*M* = 3.00, *SD* = 0.77). Only the grand mean of facilitating conditions construct was 2.78 (*SD* = 0.64), which was lower than 3.00; the other four grand means of constructs were all slightly higher than 3.0, indicating that the Tennessee respondents slightly agreed with the statements provided by the survey (Table 2).

Data collected from the UC extension system showed that, similarly to data collected from Tennessee participants, the highest-scoring construct was “Behavioral Intention” (*M* = 3.53, *SD* = 0.68); the lowest-scoring construct was “Facilitating Conditions” (*M* = 2.72, *SD* = 0.78). The grand means for all four constructs were slightly higher than 3.0, except for the construct of facilitating conditions, which had a grand mean below 3.0 (*M* = 2.72; *SD* = 0.78), indicating slight agreement among UC Extension respondents with the construct statements provided in this study (Table 3).

### 4.2. Inferential Results

#### 4.2.1. The Relationships between UTAUT Constructs

A Pearson correlation coefficient was computed to determine the linear relationship between behavioral intention and performance expectancy, between behavioral intention and effort expectancy, between behavioral intention and facilitating conditions, and between behavioral intention and social influence. The results analyzed from data collected from Tennessee participants indicated significant positive relationships between behavioral intention and other four constructs, performance expectancy (*r* = 0.80, *p* < 0.01), effort expectancy (*r* = 0.53, *p* < 0.01), facilitating conditions (*r* = 0.50, *p* < 0.01), and social influence (*r* = 0.67, *p* < 0.01; see Table 4).

The Pearson correlation coefficient results analyzed from data collected from the UC extension participants indicated significant positive relationships between behavioral intention and the other four constructs, performance expectancy (*r* = 0.68, *p* < 0.01), effort expectancy (*r* = 0.49, *p* < 0.01), facilitating conditions (*r* = 0.55, *p* < 0.01), and social influence (*r* = 0.72, *p* < 0.01; see Table 5).

#### 4.2.2. Determine the Mean Differences of Variables among Participants from Two Extension Systems

We conducted a series of independent sample *t*-tests to examine the mean difference of UTAUT variables (performance expectancy, effort expectancy, social influence, facilitating conditions, and behavioral intention) among participants from two different areas. The results of the *t*-tests showed no significant mean differences between the two groups for all variables (see Table 6).

The results in Table 1 show that the two groups of respondents who were middle-aged (35 to 54 years) and older (>55 years) with longer service made up the majority of respondents, and the number of respondents was the same. Therefore, we also conducted an independent sample *t*-test to investigate the mean difference in participants’ behavioral intentions from two different age groups with 30 participants in each group. The result of the *t*-test indicated no significant mean differences between the two groups (see Table 7).

The results in Table 1 show that the two groups of respondents with fewer than 10 years of service and those with more than 20 years of service made up the majority of respondents, and the number of respondents was similar. Therefore, we conducted an independent sample *t*-test to investigate the mean difference in participants’ behavioral intentions from two different years of service groups. The result of the *t*-test indicated no significant mean differences between the two groups (Table 8).

The results in Table 1 showed that more than 69% of respondents were male. Therefore, we conducted chi-squared analysis, and the results indicated that each independent variable—including performance expectancy (*X*^2^(4, *N* = 51) = 37.12, *p* < 0.01), effort expectancy (*X*^2^(4, *N* = 51) = 17.71, *p* < 0.01), social influence (*X*^2^(4, *N* = 51) = 19.83, *p* < 0.01), and promotion condition (*X*^2^(4, *N* = 51) = 14.19, *p* = 0.01)—had a significant relationship with the dependent variable, the behavioral intention of extension professionals to promote precision agriculture (see Table 9).

#### 4.2.3. Determine the Predictor of Behavioral Intention to Promote Precision-Farming Technologies

For the final objective of this study, we sought to determine whether the seven independent variables (performance expectancy, effort expectancy, social influence, facilitating conditions, age, gender, and years of service) used in this study could be predictors of extension professionals’ behavioral intention to promote precision-farming technologies. To assess the multiple linear regression analysis, we coded four UTAUT predictors, performance expectancy, effort expectancy, social influence, and facilitating conditions as 1 = *strongly disagree*, 2 = *disagree*, 3 = *neither*, 4 = *agree*, to 5 = *strongly agree*. In addition, the moderator gender was coded as 1 = *male*, 2 = *female*, 3 = *other*, and 4 = *rather not answer*. The independent variable, age, was analyzed using the actual age of the participants as a quantitative variable rather than a categorical variable in this study to include more information.

We regressed the dependent variable, behavioral intention to promote precision agriculture technologies, on the independent variables. The table below illustrates the multiple linear regression model (R^2^ = 0.65, *F* = 15.67, *p* = < 0.01). The model explained 65% of the variance in behavioral intention scores. The results indicated that two constructs, performance expectancy (*p* < 0.01) and social influence (*p* < 0.01), significantly predicted behavioral intention. In addition, the other two constructs, effort expectancy (*p* = 0.58) and facilitating conditions (*p* = 0.07), were not significant predictors of behavioral intention. The regression analysis results also revealed no statistically significant interaction between the effects of moderators of age, gender, and years of service on individuals’ behavioral intention to promote precision farming technologies. Participants predicted behavioral intention to promote precision farming technologies was equal to 0.33 (intercept) + 0.36 performance expectancy + 0.36 social influence. The regression model explained that 65% of the variance of extension professionals’ intention to promote the technologies was due to their beliefs of performance expectancy and the social influence of precision farming technologies for agriculturalists (Table 10).

## 5. Discussions

The descriptive results revealed that the grand means for behavioral intention, performance expectancy, effort expectancy, and social influence indicated that extension professionals in both states slightly agreed with the statements provided by the instrument used in this study. However, the grand means of facilitating conditions in both states indicated that extension professionals slightly disagreed with the statements provided by the instrument, indicating that extension professionals believed that the lack of suitable technical and organizational environmental support would reduce their intention to promote precision agriculture.

According to the findings of this study, there were no significant differences between UT and UC extension participants. Age and years of service were not variables that affected extension professionals’ intentions to promote precision-farming technologies. In addition, older or younger extension professionals and those of different amounts of service experience did not significantly affect the extension professionals’ behavioral intention to promote precision farming technologies. The professionals surveyed are willing to promote farming technologies as long as they have opportunities to enhance their performance expectancy, effort expectancy, social influence, and facilitating conditions.

The regression results are consistent with the previous studies [50,51] applying the UTAUT model to the field of technology use, indicating that performance expectancy and social influence are significant predictors of technology acceptance. The finding indicates that an extension professional’s or others’ perceived expectation that an extension professional will benefit from improved performance may be a motivator for promoting the technology adoption.

## 6. Conclusions and Implications

The data revealed gaps in information dissemination, professional development needs for current and future professionals’ training opportunities, and strategies for improving farmer adoption. The adoption of precision farming technologies is one strategy for improving sustainable food chains and systems. Agricultural extension, change agents, and professionals serve essential roles in disseminating innovation attributes across local communities in the battle to sustain food chains and improve land sustainability. Given the low response rate and focus on two states, we recognize that the data can be generalized to only the two state extension systems investigated. However, findings can be used by others to guide future studies.

The very strong and substantial significant positive correlation coefficients of all UTAUT independent variables on behavioral intention indicated that extension professionals need to be supported with professional development training to enhance their competencies in promoting precision agriculture technologies. Extension professionals in both state extension systems had similar intentions to promote precision agriculture technologies to stakeholders [29]. The data informs extension administrators, program leaders, professional development staff, and specialists of variables necessary in professional development to improve extension professionals’ promotion of precision farming technologies.

The results indicated that there were no significant differences in the agreements of all constructs (performance expectancy, effort expectancy, social influence, facilitating conditions, and behavioral intention) used in this study between extension professionals from two state extension systems. In addition, gender, age, and years of service were not variables that affected extension professionals’ intention to promote precision farming technologies. Thus, we concluded that there is no need to target a specific age, gender, or years of service experience range and that we should instead use a generalized approach. Data indicated that extension professionals believed their behavior to promote precision farming technologies was related to performance expectancy, effort expectancy, social influence, and facilitating conditions. The regression model explained that 65% of the variance of extension professionals’ intention to promote the technologies was due to their beliefs of performance expectancy and the social influence of precision agriculture technologies for agriculturalists.

Many studies have examined the implementation of the UTAUT model with the technology’s acceptance [52]. Still, few research appear to have explored the acceptance of the promotion of precision agriculture from the agricultural extension professionals’ perspectives. Therefore, this study has expanded the UTAUT application in an agriculture context with new technology (precision farming).

The current study provides a primary understanding of factors influencing agricultural extension professionals’ behavioral intention to promote precision farming technology adoption. We further suggest training programs addressing the development of advanced competencies needed to promote agricultural innovation as supported by [26]. Agricultural program development specialists must be aware that age, gender, or years of service are not barriers to extension agents and specialists promoting precision farming technologies. The results of performance expectancy, effort expectancy, social influence, facilitating conditions, and behavioral intention [29] among extension agents and specialists from different states highlight the importance of how training programs can comprehensively apply to different areas to help effectively facilitate the diffusion of precision agriculture technologies and bridge the gap in precision agriculture information dissemination channels. Social network systems, such as mentorship, can be leveraged to encourage extension professionals to engage in precision agriculture dissemination. This study contributes to the future professional development programs for agricultural extension professionals on communicating agricultural innovations with educators, extension agents and specialists, producers, and stakeholders to address food security, climate change adaptation, and sustainability issues [21].

## Figures and Tables

**Figure 1 foods-12-02208-f001:**
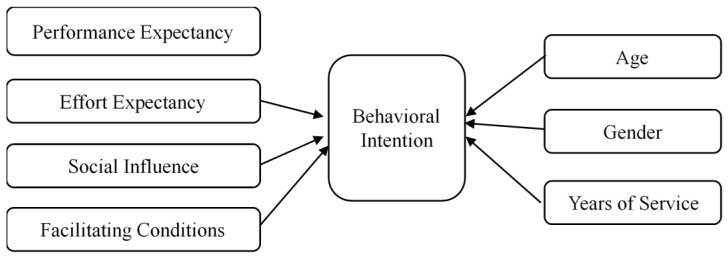
Modified Venkatesh et al.’s [29] UTAUT model used in this study.

**Figure 2 foods-12-02208-f002:**
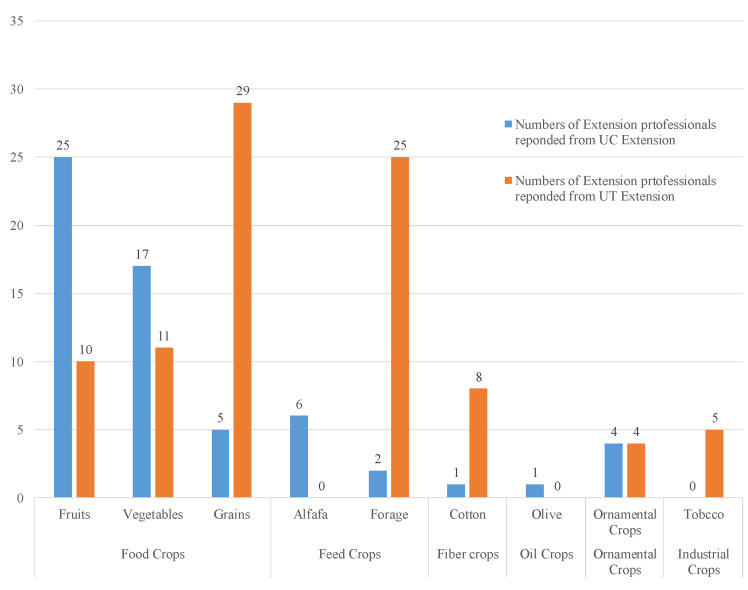
Crop variety worked on by surveyed extension professionals in two extension systems.

**Table 1 foods-12-02208-t001:** Participants’ demographic characteristics.

Characteristics	*f*			%
	UT Extension	UC Extension	Total	
Gender				
Male	34	17	51	68.92
Female	8	12	20	27.03
Prefer not to answer	0	3	3	4.05
				100.00
Age				
Under 34 years	7	5	12	16.67
35 to 54 years	15	15	30	41.67
55 years and older	13	17	30	41.67
				100.01
Years of Service as an Extension Agent/Specialist				
Less than ten years	12	20	32	44.44
10–20 years	7	2	9	12.50
More than 20 years	18	13	31	43.06
				100.00

**Table 2 foods-12-02208-t002:** Descriptive statistics for UTAUT constructs of UT extension participants.

Constructs	*n*	*M*	*SD*
Behavioral Intention	44	3.36	0.72
Performance Expectancy	45	3.33	0.75
Social Influence	44	3.19	0.56
Effort Expectancy	45	3.00	0.77
Facilitating Conditions	44	2.78	0.64

Note. Grand mean = 3.13, *SD* = 0.69, scale: 1 = *strongly disagree*, 2 = *disagree*, 3 = *neither*, 4 = *agree*, and 5 = *strongly agree.*

**Table 3 foods-12-02208-t003:** Descriptive statistics for UTAUT constructs of UC extension participants.

Constructs	*n*	*M*	*SD*
Behavioral Intention	37	3.53	0.68
Performance Expectancy	38	3.34	0.87
Social Influence	37	3.20	0.70
Effort Expectancy	37	3.00	0.78
Facilitating Conditions	37	2.72	0.78

Note. Grand mean = 3.16, *SD* = 0.76, scale: 1 = *strongly disagree*, 2 = *disagree*, 3 = *neither*, 4 = *agree*, and 5 = *strongly agree*.

**Table 4 foods-12-02208-t004:** Correlation between UTAUT constructs of UT extension participants.

	Performance Expectancy	Behavioral Intention	Social Influence	Effort Expectancy	Facilitating Conditions
Performance Expectancy	-				
Behavioral Intention	0.80 *	-			
Social Influence	0.64 *	0.67 *	-		
Effort Expectancy	0.59 *	0.53 *	0.33 *	-	
Facilitating Conditions	0.43 *	0.50 *	0.35 *	0.62 *	-

Note. * *p* < 0.05. Magnitude: 0.01 ≥ *r* ≥ 0.09 = *Negligible*, 0.10 ≥ *r* ≥ 0.29 = *Low*, 0.30 ≥ *r* ≥ 0.49 = *Moderate*, 0.50 ≥ *r* ≥ 0.69 = *Substantial*, *r* ≥ 0.70 = *Very Strong* (Davis, 1971).

**Table 5 foods-12-02208-t005:** Correlations between UTAUT constructs of UC extension participants.

	Performance Expectancy	Behavioral Intention	Social Influence	Effort Expectancy	Facilitating Conditions
Performance Expectancy	-				
Behavioral Intention	0.68 *	-			
Social Influence	0.58 *	0.72 *	-		
Effort Expectancy	0.48 *	0.49 *	0.51 *	-	
Facilitating Conditions	0.38 *	0.55 *	0.53 *	0.67 *	-

Note. * *p* < 0.05. Magnitude: 0.01 ≥ *r* ≥ 0.09 = *Negligible*, 0.10 ≥ *r* ≥ 0.29 = *Low*, 0.30 ≥ *r* ≥ 0.49 = *Moderate*, 0.50 ≥ *r* ≥ 0.69 = *Substantial*, *r* ≥ 0.70 = *Very Strong* (Davis, 1971).

**Table 6 foods-12-02208-t006:** *t*-test results for comparing UT extension participants and UC extension participants.

Group	*n*	*M*	*SD*	*t*	*df*	*p*
Performance Expectancy						
UC Extension	38	3.34	0.75	−0.04	81	0.97
UT Extension	45	3.33	0.87			
Effort Expectancy						
UT Extension	45	3.00	0.77	0.07	80	0.95
UC Extension	37	2.99	0.78			
Social Influence						
UC Extension	37	3.20	0.56	−0.06	79	0.95
UT Extension	44	3.19	0.70			
Facilitating Conditions						
UT Extension	44	2.78	0.64	0.35	79	0.73
UC Extension	37	2.72	0.78			

Note. Scale: 1 = strongly disagree, 2 = disagree, 3 = neither, 4 = agree, and 5 = strongly agree.

**Table 7 foods-12-02208-t007:** *t*-test results for comparing participants’ age Groups.

Group	*n*	*M*	*SD*	*t*	*df*	*p*
Behavioral Intention						
55 years and older	30	3.46	0.75	−1.12	58	0.27
35 to 54 years	30	3.26	0.67			

Note. Scale: 1 = strongly disagree, 2 = disagree, 3 = neither, 4 = agree, and 5 = strongly agree.

**Table 8 foods-12-02208-t008:** *t*-test results for comparing participants’ years of service groups.

Group	*n*	*M*	*SD*	*t*	*df*	*p*
Behavioral Intention						
Fewer than 10 years of service	32	3.72	0.57	1.64	61	0.11
More than 20 years of service	31	3.47	0.65			

Note. Scale: 1 = strongly disagree, 2 = disagree, 3 = neither, 4 = agree, and 5 = strongly agree.

**Table 9 foods-12-02208-t009:** A comparison of the chi-squared for four UTAUT constructs and behavioral intention construct in male participants.

	Negative Intention		Moderate Intention		Positive Intention			
	*n*	%	*n*	%	*n*	%	*X* ^2^	*p*
Performance Expectancy								
Negative Opinion	6	11.8	3	5.9	0	0	37.12	<0.01 *
Moderate Opinion	5	9.8	17	33.3	4	7.8		
Positive Opinion	0	0	2	3.9	14	27.5		
Effort Expectancy								
Negative Opinion	7	13.7	11	21.6	1	2.0	17.71	<0.01 *
Moderate Opinion	3	5.9	11	21.6	11	21.6		
Positive Opinion	1	2.0	0	0	6	11.8		
Facilitating Conditions								
Negative Opinion	9	14	14	27.5	4	7.8	14.19	0.01 *
Moderate Opinion	2	8	8	15.7	11	21.6		
Positive Opinion	0	0	0	0	3	5.9		
Social Influence								
Negative Opinion	5	4	4	7.8	0	0	19.83	<0.01 *
Moderate Opinion	6	18	18	35.3	12	23.5		
Positive Opinion	0	0	0	0	6	11.8		

Note. * *p* < 0.05. Three groups were used for each independent variable, coded as Negative Opinion = scale 1–2, Moderate Opinion = scale 3, and Positive Intention = 4–5. Three groups were used for Behavioral Intention, coded as Negative intention = scale 1–2, Moderate intention = scale 3, and Positive intention = scale 4–5. Scale: 1 = strongly disagree, 2 = disagree, 3 = neither, 4 = agree, and 5 = strongly agree.

**Table 10 foods-12-02208-t010:** Regression coefficients of all independent variables on behavioral intention.

Independent Variable	Beta	*SE*	*ꞵ*	*t*	*p*
Performance Expectancy	0.36	0.10	0.42	3.75	<0.01 *
Social Influence	0.36	0.11	0.32	3.21	<0.01 *
Facilitating Conditions	0.18	0.10	0.20	1.84	0.07
Effort Expectancy	0.05	0.09	0.06	0.55	0.58
Gender	0.01	0.08	0.01	0.07	0.94
Age	0.01	0.01	0.15	0.90	0.37
Years of Service	−0.01	0.01	−0.23	−1.39	0.17

Note. * *p* < 0.05.

## Data Availability

Data is contained within the article.

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
