# Peer review of "A Correlational Study of Two U.S. State Extension Professionals’ Behavioral Intentions to Improve Sustainable Food Chains through Precision Farming Practices"

_foods, 2023, doi:10.3390/foods12112208_

Round 1

Reviewer 1 Report

Dear Authors,

Please find below my suggestions on how to improve the ms.

Why is Extension capitalized?

In general, it’s not clear to me why you emphasise so much the issue of food security, while there is no theoretical or methodological link to it. Also structuring the statement of the problem (introduction and abstract) towards food security is misleading. 

Line 60-61: this is not clear to me

Lines 79-84: this is the objective of the study. However, the methods, unit of analysis and rationale behind their selection are not mentioned. More importantly, the contribution of the study is missing.

A literature review section about related research is needed to support the interpretation of study findings

In my opinion, the methods section should be rearranged, by merging the theoretic al framework section with the data analysis subsection into a “methodology” section to be followed by a “Data” section (current sections 3.1+3.2). Methodological explanation that is currently in the results section should be added to the new methodology section. Statistical analyses should be explained, including formulas and coefficient description for regressions.

Results and discussion: I don’t understand how findings are discussed here. I suggest adding a discussion section, where you address the limitations of the study as well

Lines 430-433: this is a very general statement. Also, why do you need a citation if it is the contribution of your research? The contribution should be linked to research gaps and presented in the introduction. In the conclusions, I think you should add some research and policy recommendations.

Author Response

Thank you for the time and detail in providing us high-quality feedback. We have attached our responses for this reviewer. 

Reviewer 2 Report

This is a good paper on an important and under-researched topic. The paper applies an appropriate model, has collected a sound and original data set, and the presentation of results and analysis is well done. Overall, the paper will make a good contribution to the journal, that will be of interest to the readers.  I would suggest considering the following points:

-Provide a definition of extension professional early in the paper. The concept will not be familiar to all readers;

-In some early parts of the paper, it reads as though technology provides solutions alone, which seems to be a determinist argument. Later on, as the paper demonstrates, people and context are critical to the development, implementation and outcomes of technology innovation. It would be worth mentioning early on that, while technology is important, so too are the people. This then also provides a further justification for the research;

-Line 61 refers to 'cultural structures of farmers.' This concept requires another phrase or sentence to explain what it means;

-Line 108, 'proven' is too strong, perhaps 'demonstrated'

-Line 121: title of figure, does not need words 'used in this study'

-Section 3.1: The names of the two extension systems are presented. Was ethics clearance provided for naming the systems?  If yes, make this clear in the text.  If not, then use pseudonyms.

-Page 147: The sampling process is presented as being a random sampling process, but it is not clear that this is the sampling process that was used. As there is no contact list, it appears that the researchers identified the population for the study based on knowledge. This appears to be a purposive sampling approach.  If it is a random sampling approach, please provide some more detail to justify this claim. In addition, is the number of 102 participants the sampling frame or the sample?

-If possible, it would be helpful to include the questionnaire as an appendix to the paper. The questionnaire is a contribution of the paper, and would also assist future researchers seeking to explore the findings of this paper in other contexts;

-Line 191, the paper says, 'a total of 102 participants accessed the instrument...'  Does this mean that 102 participants completed the questionnaire and are now included in the study?  Consider writing 'completed' (if that is the case) rather than 'accessed'.  Some people may access but not complete the questionnaire

-Line 192 indicates that 78 respondents gave their gender, whereas Table 1 seems to indicate 71 (51 Male and 20 Female).  Check tables for consistency.  Similarly, in the same sentence, it is written that 'indicating more than half were male.'  Does this mean more than half of the total number of respondents, or more than half of the participants who answered this question.  Please clarify this point, and for the other demographic data discussed in this paragraph.

-On line 390, it is indicated that results can only be generalised to the 2 extension programs investigated.  Please confirm that the sampling process was a random sampling process, to support this statement.

-On line 418, the authors write that the paper has expanded the UTAUT model.  This would be a good spot to directly revisit Figure 1, and to explain further how the research supports this  Figure. This then becomes an important contribution of the paper, for other researchers to adopt and test further.  This would also help to further develop the discussion and conclusion section of the paper, in terms of highlighting the contribution of the paper.

-Line 420, 'determining' is too strong, perhaps 'influencing'

The paper is well written for the most part.  A final proof reading will help identify sentences and phrases that need attention, in particular in terms of grammar.  For example:

Line 139-140: Sentence reads as: 'Since the technology that can be applied to precision-farming varies by crops and region.' This does not read as a sentence.  Can delete the first word 'Since'

Line 147: The written expression in the following sentence needs attention: 'To accomplish the research objectives of this study, we used a random sampling to equal select participants from the population.'

Proof read the paper for similar written expression issues throughout.

Author Response

We greatly appreciate the time and the quality of review this reviewer provided us in making our manuscript better. We have attached our responses to this reviewer. 
